# Alterations in the Gut Microbiome in the Progression of Cirrhosis to Hepatocellular Carcinoma

Yelena Lapidot,[a,b,c] Amnon Amir,[d] Rita Nosenko,[e] Atara Uzan-Yulzari,[e] Ella Veitsman,[b,f] Oranit Cohen-Ezra,[b] Yana Davidov,[b] Peretz Weiss,[b] Tanya Bradichevski,[b] Shlomo Segev,[g] Omry Koren,[e] Michal Safran,[a,b] Ziv Ben-Ari[a,b,c]

aLiver Research Laboratory, Sheba Medical Center, Tel Hashomer, Israel
bLiver Diseases Center, Sheba Medical Center, Tel Hashomer, Israel
cThe Sackler School of Medicine, Tel Aviv University, Tel Aviv, Israel
dCancer Research Center, Sheba Medical Center, Tel Hashomer, Israel
eFaculty of Medicine, Bar-Ilan University, Safed, Israel
fLiver Diseases Center, Rambam Health Care Campus, Haifa, Israel
gMedical Screening Unit, Sheba Medical Center, Tel Hashomer, Israel

Michal Safran and Ziv Ben-Ari contributed equally to this article.

**ABSTRACT** Hepatocellular carcinoma (HCC) is the second leading cause of cancer-related mortality worldwide. While cirrhosis is the main risk factor for HCC, the factors influencing progression from cirrhosis to HCC remain largely unknown. Gut microbiota plays a key role in liver diseases; however, its association with HCC remains elusive. This study aimed to elucidate microbial differences between patients with HCC-associated cirrhosis (HCC-cirrhosis) and cirrhotic patients without HCC and healthy volunteers and to explore the associations between diet, lifestyle, and the microbiome of these patients. Fecal samples and food frequency questionnaires were collected from 95 individuals (30 HCC-cirrhosis patients, 38 cirrhotic patients without HCC, and 27 age- and body mass index [BMI]-matched healthy volunteers). 16S rRNA gene sequencing was performed. Bacterial richness in cirrhosis and HCC-cirrhosis patients was significantly lower than in healthy controls. The HCC-cirrhosis group was successfully classified with an area under the curve (AUC) value of 0.9 based on the dysbiotic fecal microbial signature. The HCC-cirrhosis group had a significant overrepresentation of *Clostridium* and *CF231* and reduced *Alphaproteobacteria* abundance compared to cirrhotic patients without HCC. Patients with HCC-cirrhosis who were overweight displayed significantly decreased bacterial richness and altered microbiota composition compared to their normal-weight counterparts. There was a significant correlation in the HCC-cirrhosis group between intake of artificial sweeteners and the presence of *Akkermansia muciniphila*. A unique microbial signature was observed in patients with HCC-cirrhosis, irrespective of cirrhosis stage, diet, or treatment. BMI, dietary sugar, and artificial sweeteners were significantly associated with alterations in the microbiome of HCC-cirrhosis patients. However, the increased abundance of *Clostridium* and *CF231* observed in HCC-cirrhosis patients was not influenced by environmental factors, implying that this change was due to development of HCC.

**IMPORTANCE** Development of hepatocellular carcinoma in patients with cirrhosis is associated with alterations in intestinal microbiota, including an escalation of dysbiosis and reduced bacterial richness. This study demonstrates that reduced bacterial richness and dysbiosis escalate with the progression of cirrhosis from compensated to decompensated cirrhosis and to HCC-associated cirrhosis (HCC-cirrhosis). Moreover, we report for the first time the effect of environmental factors on HCC-cirrhosis. Excess weight was associated with increased dysbiosis in patients with HCC

Address correspondence to Yelena Lapidot, Lena.lapidot@gmail.com.

compared to their normal-weight counterparts. Moreover, fatty liver, consumption of artificial sweeteners, and high-sugar foods were associated with altered microbial composition, including altered levels of *Akkermansia muciniphila* in HCC-cirrhosis. We have successfully determined that levels of *Alphaproteobacteria* and the two genera *CF231* and *Clostridium* are significantly altered in cirrhotic patients who develop hepatocellular carcinoma, independently of cirrhosis severity and dietary habits.

**KEYWORDS** diet, *A. muciniphila*, cirrhosis, hepatocellular carcinoma, microbiome, gut microbiome

Liver cancer is currently the third leading cause of cancer-related mortality worldwide, with an increasing annual incidence and poor prognosis (1). Hepatocellular carcinoma (HCC) accounts for 85% to 90% of primary liver cancers. Cirrhosis, regardless of etiology, is the most important risk factor for the development of HCC; however, the factors influencing disease progression from cirrhosis to HCC remain largely elusive (2).

Accumulating evidence indicates that the gut microbiome has an important role in the development of liver cancer. Studies based on animal models showed that increased bacterial lipopolysaccharide levels in the cirrhotic liver activate Toll-like receptor 4 in hepatic stellate cells (HSCs) and hepatocytes, resulting in fibrogenesis and secretion of the epiregulin growth factor, which triggers tumor proliferation (3). Moreover, obesity and high-fat diet have been identified as major risk factors for HCC. Obesity has also been associated with altered gut microbiota, translocation of gut-derived bacterial products to the liver, and increased conversion of bile acids to deoxycholic acid (DCA) by pathogenic bacteria. This activates a senescence-associated secretory phenotype (SASP) in HSCs that promotes the development of HCC (4, 5). Apart from studies based on animal models, a recent study that focused on patients with nonalcoholic fatty liver disease (NAFLD)-related HCC demonstrated an increase in the abundances of *Bacteroides* and members of the family *Ruminococcaceae* and a reduced abundance of *Bifidobacterium* (6). A case-control study by Ren et al. (7) led to their reporting that fecal microbial diversity in Chinese examinees decreased from healthy controls (HC) to cirrhosis patients but increased from cirrhosis patients to HCC-associated cirrhosis (HCC-cirrhosis) patients. Abundances of butyrate-producing bacterial genera were decreased in HCC patients versus controls whereas abundances of pathogen-producing genera were increased. This report strengthens the hypothesis that there are significant microbial alterations in HCC-cirrhosis patients.

The altered microbiome may be a cause or a consequence of disease or, more likely, an environmental risk factor or disease modulator. It is becoming clear that the microbiome provides biomarkers that can be tested for risk or presence of disease (8). In this study, we aimed to characterize the alterations of the gut bacterial diversity and composition with the progression of cirrhosis to HCC-cirrhosis and, for the first time, to evaluate the associations between cirrhosis etiology, body mass index (BMI), and dietary habits and the microbiome of HCC patients. These findings might encourage the use of microbiome-oriented noninvasive diagnostics and therapeutic modalities to improve management of liver diseases.

## RESULTS

**Demographic data.** Of the 95 participants in this study, 38 were diagnosed with cirrhosis, 30 were diagnosed with HCC-cirrhosis, and 27 were age- and BMI-matched healthy controls. Cirrhosis etiologies were NAFLD and HCV, for both the cirrhosis-only and HCC-cirrhosis groups. There were no significant differences in cirrhosis etiology, severity, mean age, and BMI between the groups (Table 1).

**Patients with cirrhosis and HCC-cirrhosis display decreased species richness and altered community composition compared to healthy controls.** The overall fecal microbial diversity of both patient groups was significantly different from that of healthy controls (Fig. 1A and B), as evidenced by significantly reduced richness and

mSystems®

**TABLE 1** Demographic and clinical characterization of study groups[a]

| Parameter | Characteristic or result | Value(s) for indicated group | | | P value |
| --- | --- | --- | --- | --- | --- |
| | | Cirrhosis (n = 38) | HCC-cirrhosis (n = 30) | Healthy controls (n = 27) | |
| Demographic characteristics | No. (%) of males | 24 (64) | 22 (73) | 21 (77) | 0.384 |
| | Age (yrs) | 64.3 | 67.5 | 61.6 | 0.150 |
| | Body mass index (kg/m$^2$) | 28.2 | 27.3 | 25.7 | 0.741 |
| Laboratory results | Glucose (mg/dl) | 116.9 | 152.4 | | 0.131 |
| | Creatinine (mg/dl) | 0.88 | 0.84 | | 0.765 |
| | Bilirubin (mg/dl) | 1.18 | 1.47 | | 0.317 |
| | Albumin (g/dl) | 3.83 | 3.55 | | 0.060 |
| | Alkaline phosphatase (U/liter) | 121.5 | 176.6 | | 0.012 |
| | Alanine aminotransferase (U/liter) | 32.3 | 72.5 | | 0.001 |
| | Aspartate aminotransferase (U/liter) | 41.7 | 90.11 | | 0.000 |
| | Glutamyl transpeptidase (U/liter) | 112.7 | 182.95 | | 0.006 |
| | International normalized ratio (INR) | 1.15 | 1.2 | | 0.595 |
| | Alfa fetoprotein (ng/ml) | 4.79 | 3,192.59 | | 0.000 |
| | Model for end-stage liver disease (MELD) score | 9.00 | 9.90 | | 0.466 |
| Cirrhosis etiology | Hepatitis C virus | 19 | 16 | | 0.812 |
| | Nonalcoholic fatty liver | 19 | 14 | | |
| Cirrhosis severity | Compensated | 23 | 20 | | |
| | Decompensated | 15 | 6 | | 0.623 |
| | NA | | 4 | | |

[a]One-way analysis of variance was used to evaluate the difference among the three groups. Continuous variables were compared using the Wilcoxon-Mann-Whitney rank sum test for comparisons between patient groups. Fisher's exact test was used to compare categorical variables. NA, not applicable.

altered bacterial composition in the cirrhosis group compared to the controls (observed features P value = 0.014; unweighted Unifrac P value = 0.004) and in the HCC-cirrhosis group compared to the controls (P value = 0.028; unweighted Unifrac P value = 0.016). However, there was no significant difference in species richness and composition between the two patients cohorts. Weighted UniFrac analysis results were less sensitive in revealing significant alterations between study groups (see Fig. S1 in the supplemental material).

Evaluation of disease etiology in the cirrhosis (Fig. 1C and D) and HCC-cirrhosis (Fig. 1E and F) groups revealed significant differences in bacterial diversity (P value = 0.04) and composition (P value = 0.034) in NAFLD compared to HCV-cirrhosis patients. The relative abundances of *Ruminococcaceae* (P value = 0.048), *Lachnospiraceae* (P value = 0.034), and *Coriobacteriia* (P value = 0.036), including genus *Collinsella* (P value = 0.011), was lower in patients with NAFLD-cirrhosis than in those with HCV-cirrhosis. At the same time, the relative abundance of *Clostridiales* was higher in NAFLD-cirrhosis patients (P value = 0.043). However, these differences were not detected in comparisons of these etiology groups in HCC-cirrhosis patients.

**Significant dysbiosis in cirrhosis accelerates with disease severity.** Patients with cirrhosis demonstrated a significantly lower relative abundance of butyrate-producing bacteria, e.g., members of the families *Ruminococcaceae* (P value = 0.001) and *Lachnospiraceae* (P value = 0.047) than the controls. Moreover, members of the *Gammaproteobacteria* (P value = 0.001) and *Enterobacteria* (P value = 0.001) were significantly more abundant in cirrhotic patients. The full list of significance alterations is detailed in Table S2 in the supplemental material.

Considering the cirrhosis-only population, a significant difference in β-diversity was observed between patients with compensated versus decompensated cirrhosis (P value = 0.024) (Fig. S2A). We observed marked dysbiosis in decompensated cirrhosis, with a significantly higher abundance of several taxonomic levels of *Bacilli*, *Streptococcaceae* (P value = 0.006), *Alloscardovia* (P value = 0.031), and *Atopobium* (P value = 0.048), while *Ruminococcaceae* were depleted (P value = 0.024) (Fig. S2B

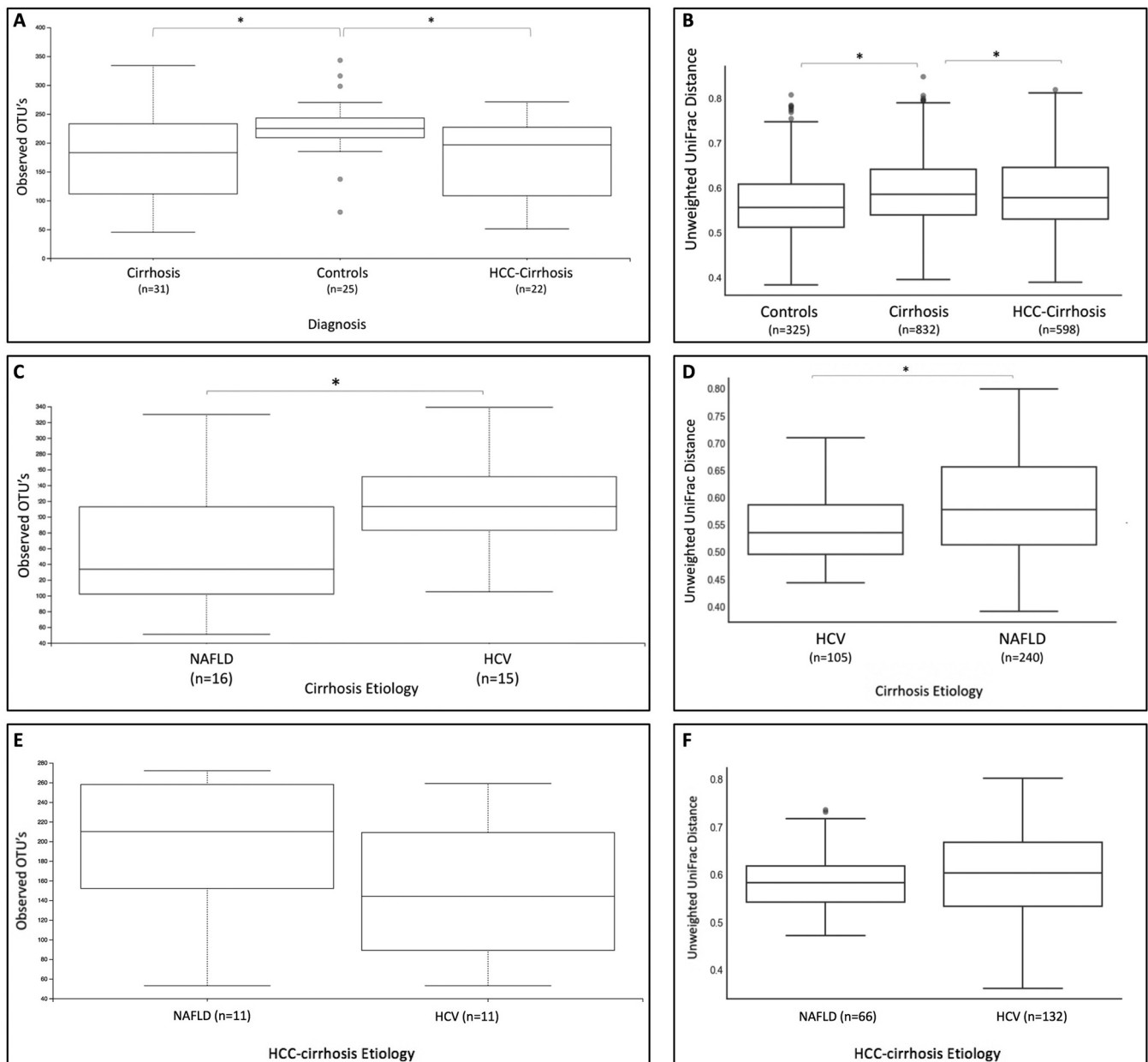

**FIG 1** Gut microbiome alterations in patients with cirrhosis or HCC-cirrhosis and in controls and cirrhosis etiology distribution in study groups. (A) Box plot of α-diversity (observed features) displaying a significant decrease in bacterial richness in cirrhotic patients and patients with HCC-cirrhosis compared to healthy controls ($P$ values = 0.014 and 0.028, correspondingly). (B) Box plot of β-diversity (unweighted UniFrac distance) displaying a significant difference in bacterial composition in cirrhotic patients and patients with HCC-cirrhosis compared to healthy controls ($P$ values = 0.004 and 0.016, correspondingly). (C and D) Box plot of α-diversity (observed OTU indices) (C) and β-diversity (unweighted UniFrac distance matrix) (D) of cirrhosis etiologies (in cirrhosis without HCC), displaying a significant decrease in bacterial richness ($P$ value = 0.04) and altered bacterial composition ($P$ value = 0.03) in cirrhotic patients with NAFLD compared to HCV-cirrhosis patients. (E and F) Box plot of α-diversity (observed OTU indices) (E) and β-diversity (unweighted UniFrac distance matrix) (F) of HCC-cirrhosis etiologies (in the HCC-cirrhosis group), showing that there were no significant differences in bacterial richness ($P$ value = 0.11) or composition ($P$ value = 0.07) in HCC patients with NAFLD-cirrhosis compared to HCV-cirrhosis.

and C). Interestingly, in the HCC-cirrhosis group, cirrhosis progression was not associated with fecal microbiome diversity (observed features $P$ value = 0.161) or composition ($P$ value = 0.143).

**Diuretics and high-protein diet are associated with altered microbiome composition in cirrhosis.** In examining cirrhosis patients, diuretics showed a significant association with altered microbiome richness and composition (observed operational taxonomic unit [OTU] $P$ value = 0.003; unweighted UniFrac $P$ value = 0.006) (Fig. S3A

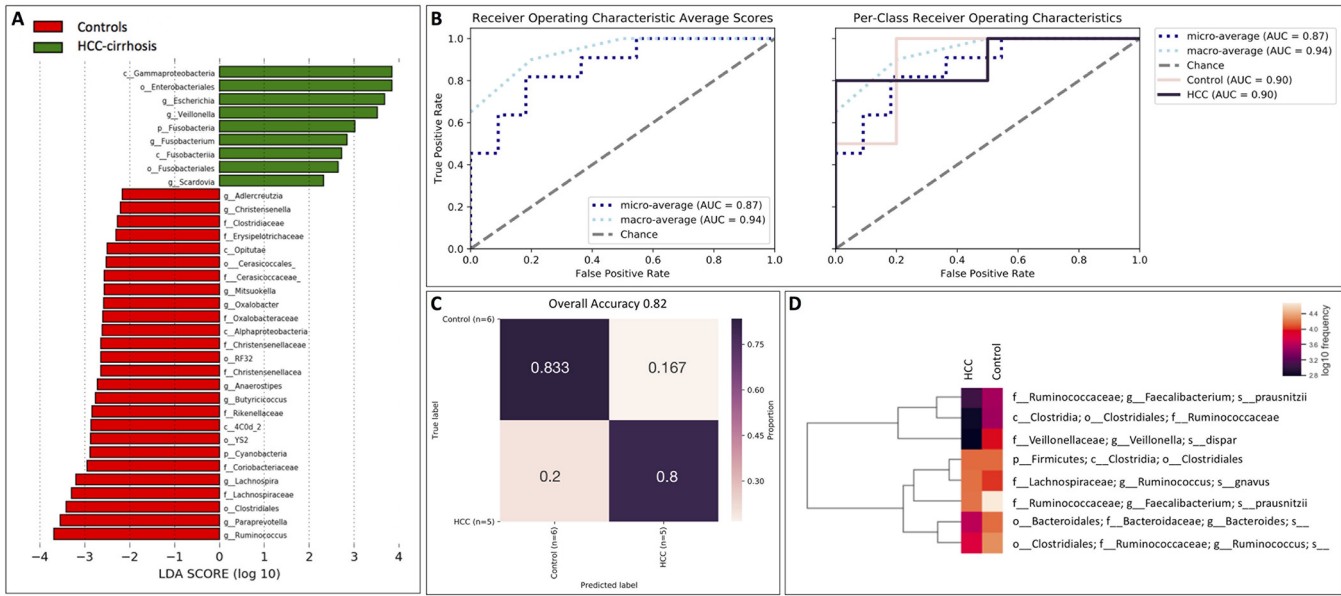

**FIG 2** Differentially abundant taxa in patients with HCC-cirrhosis compared to the controls. (A) LDA scores computed for differentially abundant taxa in the fecal microbiomes of patients with HCC-cirrhosis (green) and healthy controls (red). Length indicates effect size associated with a taxon. *P* = 0.05 for the Kruskal-Wallis test; LDA score > 2. (B) A graphical representation of the classification accuracy of a machine-learning random forest model in receiver operating characteristic (ROC) curves, displayed here as ROC curves for each class (AUC of 0.9) and average ROCs and AUCs, including "microaveraging" of 0.87 (to calculate metrics globally by averaging across each sample) and "macroaveraging" of 0.94 (to give equal weight to the classification of each sample). (C) Confusion matrix displaying the classification results, with overall accuracy of 82%, baseline accuracy of 0.545, and an accuracy ratio of 1.5. (D) Important features are represented in an abundance heat map, consisting of log₁₀ frequencies of the most important taxa in each sample and group (HCC-cirrhosis and healthy controls). These are the features that maximize model accuracy, as determined using recursive feature elimination.

and B). After adjustment for cirrhosis severity, the relative abundances of *Butyricicoccus* (*P* value = 0.007) and *Actinomyces* (*P* value = 0.001) proved higher in consumers of diuretics, with no dependence on disease progression (Fig. S3C and D). Moreover, patients with cirrhosis that reported consumption of a high-protein diet displayed alterations in bacterial composition (*P* value = 0.009) (Fig. S4A and B). Correspondingly, there were strong correlations between consumption of products with high protein content (turkey, chicken, eggs) and *Phascolarctobacterium* (Rho = 0.5, *P* value = 0.001) and between fish consumption and the genus *Anaerofilum* (Rho = 0.52, *P* value = 0.001). However, these correlations did not pass the false-discovery-rate (FDR) correction threshold of FDR < 0.05 (Table S3).

**Gut dysbiosis in HCC-cirrhosis, including increased levels of *Fusobacteriia*.** HCC-cirrhosis patients displayed a significant decrease in the relative abundance of butyrate-producing bacteria *Ruminococcaceae* (*P* value = 0.043), *Butyricicoccus* (*P* value = 0.000), and *Lachnospiraceae* (*P* value = 0.045) compared to the controls. Moreover, there was a decrease in the relative abundance of genera *Lachnospira* (*P* value = 0.039), *Anaerostipes* (*P* value = 0.004), and *Christensenella* (*P* value = 0.01). In parallel, the relative abundances of *Fusobacteriia* (*P* value = 0.012), *Gammaproteobacteria* (*P* value = 0.005), *Veillonella* (*P* value = 0.023), and *Scardovia* genus (*P* value = 0.018) were higher (Fig. 2A; see also Table S4).

**Classification of patients with HCC-cirrhosis compared to healthy controls.** A trained random forest classifier used to distinguish patients with HCC-cirrhosis compared to HC based on fecal microbiome composition (Fig. 2B; see also Fig. 2C) yielded an overall accuracy of 82% (baseline accuracy, 0.545; accuracy ratio, 1.5) with an area under the curve (AUC) value of 0.9. Among the most important features for discrimination were *Veillonella dispar*, *Faecalibacterium prausnitzii*, and *Ruminococcus gnavus* (Fig. 2D; see also Table S5).

**HCC-cirrhosis harbors a unique microbiome signature compared to cirrhosis without HCC.** We observed significant differences in bacterial composition between

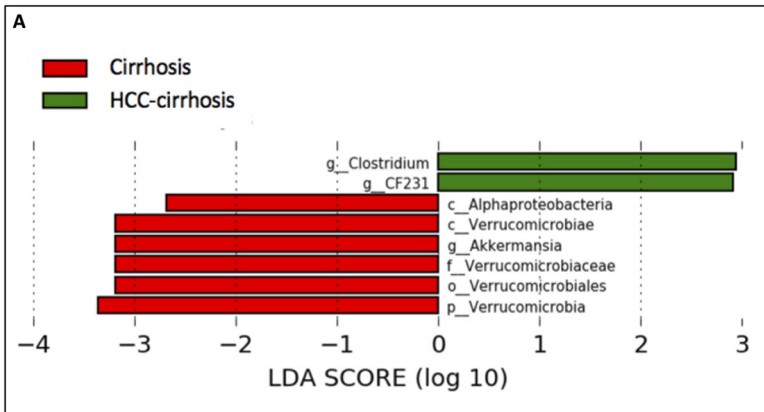
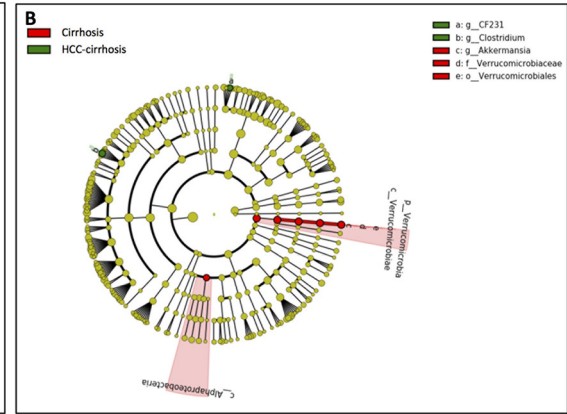

**FIG 3** Differentially abundant taxa in patients with HCC-cirrhosis compared to cirrhotic patients without HCC. (A) LDA scores computed for differentially abundant taxa in the fecal microbiomes of patients with liver cirrhosis (red) and patients with HCC-cirrhosis (green). Length indicates effect size associated with a taxon. $P = 0.05$ for the Kruskal-Wallis test; LDA score > 2. (B) Taxonomic cladogram from LEfSe showing differences in fecal taxa of cirrhosis patients compared to HCC-cirrhosis patients. There were differences in the relative abundances of *Alphaproteobacteria* (*P* value = 0.039), Clostridium (*P* value = 0.024), *CF231* (*P* value = 0.010), *Verrucomicrobia* (*P* value = 0.036), and *Akkermansia muciniphila* (*P* value = 0.039).

patients with HCC-cirrhosis and those with cirrhosis only, with an overrepresentation of *Clostridium* (*P* value = 0.024) and *CF231* genus of *Paraprevotella* (*P* value = 0.01) paralleling significantly lower abundances of *Alphaproteobacteria* (*P* value = 0.039) and *Verrucomicrobia* (in several taxonomic levels) along with *Akkermansia muciniphila* (*P* value = 0.039) in patients with HCC-cirrhosis (Fig. 3). We note that these results were conserved across cirrhosis severity and progression, i.e., decompensation and etiology did not have a significant association with these microbial alterations.

**Obesity and fatty liver are associated with significant differences in bacterial richness and composition in HCC-cirrhosis.** Patients with HCC-cirrhosis that were overweight (BMI > 25) showed significantly altered bacterial richness (Shannon's diversity *P* value = 0.024) compared to their lean counterparts (Fig. 4A). Furthermore, there was a significant difference in bacterial composition (*P* value = 0.033) (Fig. 4B), including higher relative abundance of *Campylobacter* (*P* value = 0.032), in HCC-cirrhosis patients that were overweight (Fig. 4C).

Moreover, patients with HCC-cirrhosis with fatty liver exhibited significantly lower bacterial richness than patients with HCC-cirrhosis without fatty liver (Faith's phylogenetic diversity *P* value = 0.025) and a significant difference in bacterial composition from the latter (*P* value = 0.008) (Fig. 5A and B). The significant differences included a higher relative abundance of *Verrucomicrobia*, including genus *Akkermansia* (*P* value = 0.018) (Fig. 5C).

**Artificial sweeteners and sugar consumption in HCC-cirrhosis patients significantly correlated with altered bacterial composition.** Correlation analysis of food frequency questionnaire (FFQ) data in the HCC-cirrhosis group (Table S6) revealed a strong correlation between consumption of artificial sweeteners and presence of *Akkermansia* (Rho = 0.61, *P* value = 0.002). Additionally, consumption of high-sugar foods correlated with *Cloacibacillus*, of the *Synergistetes* phylum (Rho = 0.584, *P* value = 0.003).

## DISCUSSION

The present study sought to characterize the fecal bacterial alterations in the progression of cirrhosis to HCC. Because cirrhosis, regardless of etiology, remains the most important risk factor for the development of HCC, we compared fecal microbial signatures of patients with HCC-cirrhosis to those of patients with cirrhosis without HCC and those of age and BMI-matched healthy volunteers.

Our results demonstrate that each stage of advanced chronic liver disease (compensated cirrhosis, decompensated cirrhosis, and HCC-cirrhosis) is characterized by

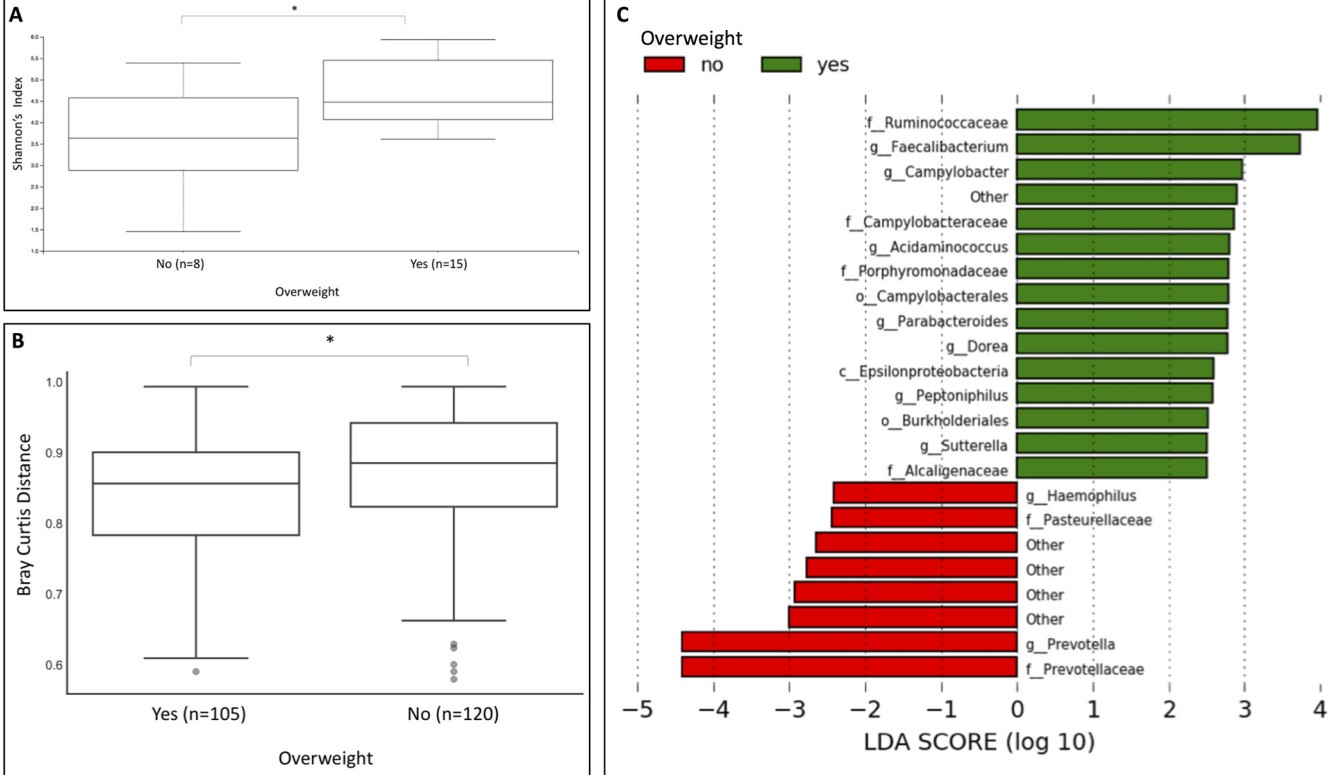

**FIG 4** Gut microbiome alterations in overweight HCC-cirrhosis patients. (A) Box plot of α-diversity (Shannon's index) displaying a significant decrease in diversity in patients with HCC-cirrhosis that were overweight (BMI > 25) compared to counterparts that were not overweight (*P* value = 0.024). (B) Box plot of β-diversity displaying a significant difference in bacterial composition in overweight patients with HCC-cirrhosis compared to counterparts that were not overweight (*P* value = 0.033). (C) LDA scores computed for differentially abundant taxa in the fecal microbiomes of overweight patients with HCC-cirrhosis. Length indicates effect size associated with a taxon. *P* = 0.05 for the Kruskal-Wallis test; LDA score > 2.

specific alterations of fecal microbial population. We demonstrated that cirrhosis and HCC-cirrhosis are associated with qualitative changes in the intestinal microbiota, including significantly lower bacterial richness and altered bacterial composition. Unweighted UniFrac identified clearer patterns of variation between samples than weighted UniFrac, implying that gut bacterial alterations occur in low-abundance features in cirrhosis and HCC-cirrhosis.

Moreover, the alterations in gut microbiota were more prominent with increasing severity of disease and were characterized by an overgrowth of potentially pathogenic, potent endotoxin-producing bacteria (i.e., members of the family *Enterobacteriaceae*, *Fusobacterium*, and *Atopobium*) and a decrease in the levels of potentially beneficial bacteria, specifically, of butyrate-producing bacteria (i.e., *Butyricicoccus*, *Anaerostipes*). These shifts may result in the decrease of anti-inflammatory short-chain fatty acids (SCFAs) and may enhance leaky gut and gut dysbiosis.

Patients with HCC-cirrhosis displayed decreased bacterial diversity and profound dysbiosis compared to the controls, with a lower abundance of potentially beneficial, butyrate-producing bacteria (i.e., *Lachnospira*, *Ruminococcus*, and *Butyricicoccus*) and an overrepresentation of pathogenic bacteria, including *Fusobacteriia*, reported to be involved in a wide spectrum of human infections and to have a role in colorectal cancer (9). Moreover, the abundance of *Veillonella* and *Scardovia* genus was higher, indicating invasion of oral bacterial species into the gut. Interestingly, *Fusobacterium*, *Veillonella*, and *Scardovia* were not significantly enriched in patients with cirrhosis compared with controls; thus, HCC-cirrhosis patients may present an escalation of bacterial translocation and dysbiosis. These results were independent from etiology or cirrhosis severity. Interestingly, Chinese patients with HCC-cirrhosis also displayed a decrease in levels of butyrate-producing bacteria (*Ruminococcus*, *Oscillibacter*, *Faecalibacterium*, and *Copro-*

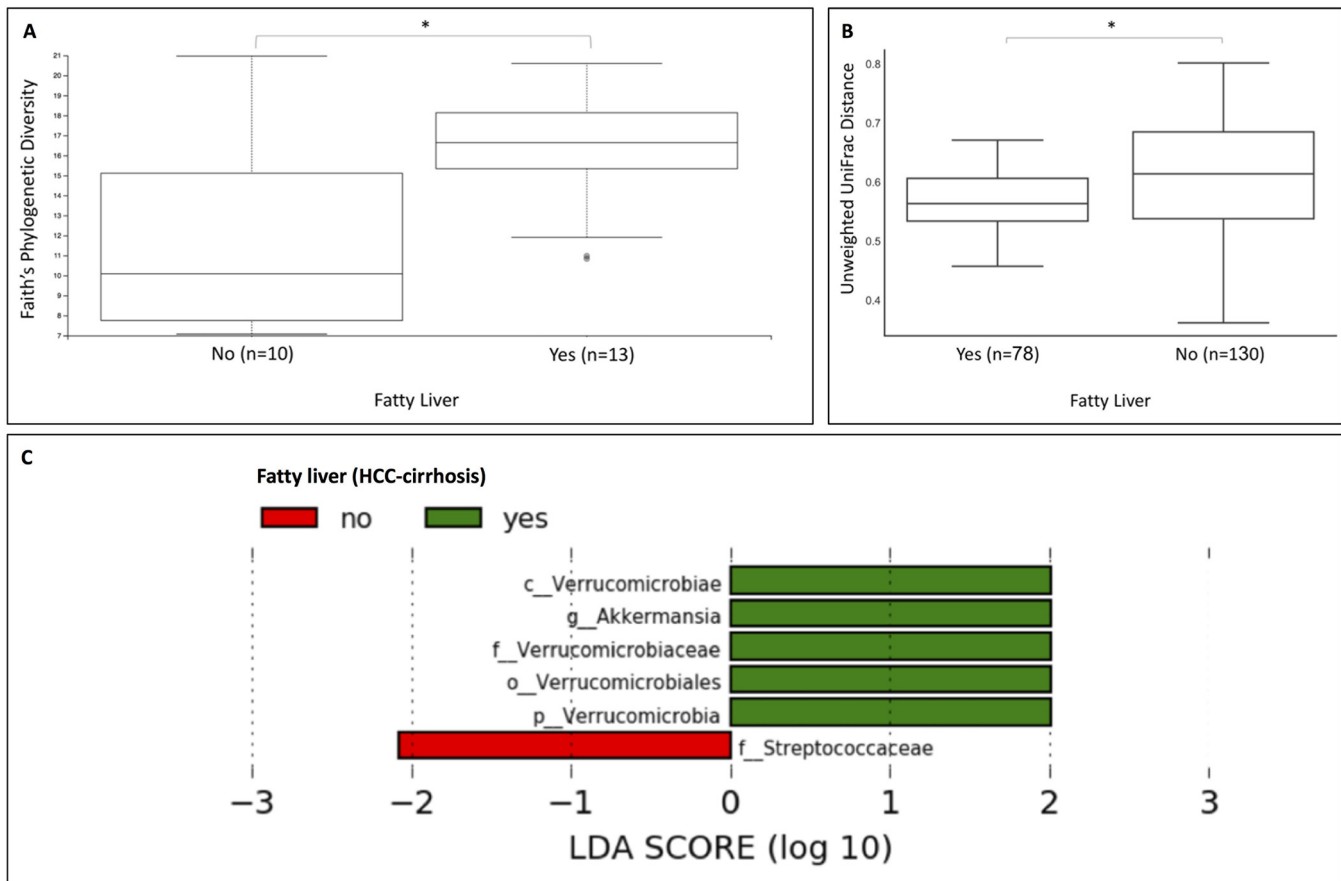

**FIG 5** Fatty liver in HCC-cirrhosis and significant association with the relative abundance of *Akkermansia*. (A) Box plot of α-diversity (Faith's phylogenetic diversity [PD]) displaying a significant decrease in diversity in patients with HCC-cirrhosis that had a fatty liver compared to counterparts without a fatty liver (*P* value = 0.025). (B) Box plot of β-diversity (unweighted UniFrac distance matrix) displaying a significant difference in bacterial composition in patients with HCC-cirrhosis that had a fatty liver compared to counterparts without a fatty liver (*P* value = 0.008). (C) LDA scores computed for differentially abundant taxa in the fecal microbiomes of patients with HCC-cirrhosis that had a fatty liver. Length indicates effect size associated with a taxon. *P* = 0.05 for the Kruskal-Wallis test; LDA score > 2.

*coccus*), while different pathogenic genera, *Klebsiella* and *Haemophilus*, were increased in abundance compared with controls (7). These results support the hypothesis that the intestinal microbiota might play a key role in cirrhosis pathogenesis and severity and, possibly, in the progression to HCC. We present a successful (AUC of 0.9) classification of patients with HCC-cirrhosis based on fecal microbiome composition, with *Veillonella* being one of the most important features for discrimination.

We found a progressive decrease in bacterial richness from healthy controls to cirrhosis and to HCC-cirrhosis patients. Interestingly, Ren et al. (7) observed such a decrease from controls to cirrhosis patients; however, there was an increase in HCC-cirrhosis patients compared to controls. These differences may have been due to geographic, demographic, or cultural and nutritional differences or may have been due to differences in the etiology and severity of cirrhosis. Moreover, Chinese patients have a unique microbiome that is influenced by geographic, demographic, cultural, and marked nutritional differences from Western populations. As previously described in a review by Fukui et al. (10), there are differences in the microbial signature of Chinese patients with cirrhosis from that of Western patients; however, Chinese patients and Western patients exhibit a common path of dysbiosis.

Microbiota comparisons between patients with liver cirrhosis and patients with HCC-cirrhosis revealed an overrepresentation of *Clostridium* and *CF231* in the HCC-cirrhosis group. *CF231* is a genus in the *Paraprevotellaceae* family, recently associated with fatty liver (11) and with higher BMI (12); both represent known risk factors for the

mSystems®

development of HCC. Moreover, we noted elevated abundance of the *Clostridium* genus belonging to the *Clostridiaceae* family, which has been shown to be correlated with high fat diets (13). Interestingly, an increase in levels of *Clostridium* cluster XI was reported in obese mice and high-fat diet mouse models of HCC, suggesting that this genus may have influenced hepatic carcinogenesis (14, 15).

Our study results reveal, similarly to Ren et al. (7), a decrease in several taxonomic levels of *Verrucomicrobia* in HCC-cirrhosis patients compared to healthy controls. However, we demonstrate that *Verrucomicrobia* levels were affected by intake of artificial sweeteners and by a state of fatty liver in these patients. Patients with fatty liver had a lower abundance of all taxonomic levels of *Verrucomicrobia*, including genus *Akkermansia muciniphila* (*A. muciniphila*). However, *Clostridium* and *CF231* were not found to be associated with factors other than HCC (e.g., diet, BMI, and fatty liver).

Interestingly, the abundance of *A. muciniphila* in humans has been reported to be lower under several pathological conditions, such as obesity, diabetes, hypertension, hypercholesterolaemia, and liver disease. Moreover, supplementation with *A. muciniphila* resulted in decreased body weight and reduction in inflammation and hepatic steatosis in animal models (16). A recent study demonstrated that ethanol exposure reduces intestinal *A. muciniphila* abundance in both mice and humans and can be recovered by oral supplementation in experimental alcoholic liver disease (ALD) models. This bacterium promotes intestinal barrier integrity and ameliorates experimental ALD, suggesting that patients with ALD might benefit from *A. muciniphila* supplementation (13).

We demonstrated that the microbiome of patients in the HCC-cirrhosis group was affected by obesity, which was associated with a further increase in the abundance of pathogenic bacteria, including *Campylobacter*. Correspondingly, animal models of HCC development singled out obesity as an independent risk factor for the development of HCC-cirrhosis; obesity itself increases the risk of HCC 1.5-fold to 4-fold. The relative risks of HCC are 117% for overweight subjects and 189% for obese patients (2).

Our findings demonstrate the potential of fecal microbes as tools for noninvasive diagnosis or microbiome-oriented interventions in HCC-cirrhosis. However, this study focused on characterization of the 16S rRNA bacterial alterations and the potential effects of etiology, cirrhosis progression, and factors such as BMI and diet on these patients' bacterial composition; thus, causality remains an open question. Nevertheless, this study showed sufficient power for detecting significant microbial alterations and successfully illuminating the effect of environmental factors on the gut microbiome of patients with HCC-cirrhosis. In the future, there will be a need for further studies of large-scale, multiracial, metagenomic cohorts that include a careful evaluation of environmental factors, including dietary habits and their effect on patients with HCC-cirrhosis. Moreover, there is a need for intervention studies, both in animal models and in humans, focusing on fecal microbiome modulation in cirrhosis and HCC-cirrhosis. These studies will allow further understanding of the roles of altered bacterial levels in the progression of cirrhosis to HCC.

## MATERIALS AND METHODS

**Subjects.** Recruitment was performed in accordance with the Helsinki Declaration and Rules of Good Clinical Practice. The study was approved by the Institutional Review Board of the Sheba Medical Center (study number 2530-15). All subjects provided written informed consent to participate in the study.

Patients were recruited from the Liver Diseases Center or the Institute for Medical Screening in Sheba Medical Center, Tel Hashomer, Ramat-Gan, Israel. The study groups included 38 patients with NAFLD ($n = 19$)-related or hepatitis C virus (HCV) ($n = 19$)-related liver cirrhosis; 30 cirrhotic subjects first diagnosed of an early stage HCC, prior to treatment (NAFLD-related HCC-cirrhosis [$n = 14$] and HCV-related HCC-cirrhosis [$n = 16$]); and 27 age- and BMI-matched healthy controls.

Cirrhosis diagnosis was based on histological and/or clinical findings (laboratory parameters, imaging findings, signs of portal hypertension at liver imaging or endoscopy) (17). Patients with HCV cirrhosis were recruited before HCV eradication. Subjects in the control group had no history of liver disease and no significant alcohol consumption. HCC diagnosis was based on imaging (triphasic computed tomography [CT] scan and magnetic resonance imaging [MRI] and/or a liver biopsy) and serum level of alpha fetoprotein (18).

The following exclusion criteria were implemented in all groups:

1. Treatment with antibiotics, probiotics, prebiotics, and laxatives during the previous 3 months.
2. Other cirrhosis etiologies, including hepatitis B virus (HBV), human immunodeficiency virus (HIV), alcoholic steatohepatitis, cholestatic disorders (primary biliary cholangitis or primary sclerosing cholangitis), and inherited liver disorders leading to cirrhosis, i.e., hemochromatosis, Wilson's disease, and alpha-1 antitrypsin deficiency.
3. A diagnosis of inflammatory bowel disease or celiac disease.
4. A diagnosis of other malignancy (not HCC) in the last 3 years.

**Sample collection.** A stool sample was collected by the participants, transported within 2 h to the laboratory, and stored at −80°C until total DNA was extracted (all fecal samples underwent a single thaw before DNA extraction). All participants completed a validated lifestyle and food frequency questionnaire (FFQ) (19–21) (see Table S1 in the supplemental material). Participants' clinical data were collected from electronic medical records.

**DNA extraction and sequencing.** Microbial DNA was extracted from fecal samples using a PureLink Microbiome DNA purification kit (Invitrogen, Thermo Fisher Scientific, Carlsbad, CA, USA), according to the manufacturer's instructions following a preliminary step of bead-beating for 2 min and elution in 50 $\mu$l elution buffer. Purified DNA was subjected to PCR amplification using PrimeSTAR Max (TaKaRa-Clontech, Shiga, Japan) for the variable V4 region (using 515F-806R barcoded primers) of the 16S rRNA gene, as previously described (22).

Amplicons were purified using Agencourt AMPure XP magnetic beads (Beckman Coulter, Brea, CA) and subsequently quantified using a Quant-It PicoGreen double-stranded DNA (dsDNA) quantitation kit (Invitrogen, Carlsbad, CA). Equimolar amounts of DNA from individual samples were pooled, cleaned by the use of E-gel (Life Technologies, Carlsbad, CA, USA), and sequenced using the Illumina MiSeq platform at the Genomic Center of the Bar-Ilan University, at the Azrieli Faculty of Medicine.

**16S rRNA gene sequencing and statistical analysis.** The sequences were analyzed using QIIME2 software packages (23). Deblur was used for sequence quality control and feature table construction, with a trim length of 150 (24, 25). Taxonomy was assigned using the QIIME2 RDP classifier algorithm, at 99% identity to the Greengenes 13.8 reference database (26).

For phylogenetic-tree-based analyses, each feature was represented by a single sequence that was aligned using the mafft program (27). A phylogenetic tree was built with Fast-Tree (28) and used to estimate the phylogenetic distances between features. $\alpha$-Diversity (Faith's phylogenetic diversity [29], observed features [30], Shannon diversity index [31], and evenness Pielou's index [32]) and $\beta$-diversity (unweighted UniFrac [33] and weighted UniFrac [34]) values were calculated using QIIME2 core-metrics-phylogenetic method, at sampling depth of 10,000.

Differential abundance was estimated using the linear discriminant analysis (LDA) effect size (LEfSe) method, which emphasizes both statistical significance and biological relevance. The algorithm performs a nonparametric factorial Kruskal-Wallis sum rank test and LDA to determine statistically significant different features among taxa and estimates the effect size of the difference (35). Differences were considered significant for $P$ values of <0.05 and a logarithmic LDA score cutoff of ≥2.

Random forest (RF) models were generated using qiime2 q2-sample-classifier, with 700 trees, optimized feature selection using recursive feature elimination, and automatically tuning hyperparameters using random grid search. Cross-validation (20-fold) was performed by the use of repeated stratified K Fold. The training was performed on 80% of the samples, and the test size was 20% of the samples.

Descriptive statistical analyses and Spearman correlation analysis were performed using R, version 3.4.4, with the package corrplot (36). Fisher's exact test was used to compare categorical variables and the Kruskal-Wallis test for continuous and categorical variables. Multiple-testing correction was performed whenever applicable using FDR (37); adjusted $P$ values of <0.05 were considered significant.

## SUPPLEMENTAL MATERIAL

Supplemental material is available online only.

**FIG S1**, TIF file, 1 MB.
**FIG S2**, TIF file, 1.9 MB.
**FIG S3**, TIF file, 1.9 MB.
**FIG S4**, TIF file, 1.9 MB.
**TABLE S1**, DOCX file, 0.02 MB.
**TABLE S2**, DOCX file, 0.01 MB.
**TABLE S3**, DOCX file, 0.01 MB.
**TABLE S4**, DOCX file, 0.01 MB.
**TABLE S5**, DOCX file, 0.01 MB.
**TABLE S6**, DOCX file, 0.01 MB.

## ACKNOWLEDGMENTS

We thank Uri Gophna.

We declare that there are no competing interests.

We contributed to the study as follows: Y.L., study concept and design, acquisition

of data, analysis and interpretation of data, drafting of the manuscript, and statistical analysis; A.A., analysis and interpretation of data and statistical analysis; R.N., analysis and interpretation of data; A.U.-Y., analysis and interpretation of data; E.W., acquisition of data; O.C.-E., acquisition of data; Y.D., acquisition of data; P.W., acquisition of data; T.B., acquisition of data; S.S., acquisition of data; O.K., critical revision of the manuscript for important intellectual content; M.S., study concept and design and study supervision; Z.B.-A., study concept and design and study supervision.

All of us had access to the study data and reviewed and approved the final manuscript.

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
