## [Reviewer comments · mSystems]

Alterations in the gut microbiome in the progression of cirrhosis to hepatocellular carcinoma

Yelena Lapidot, Amnon Amir, Rita Nosenko, Atara Uzan-Yulzari, Ella Veitsman, Oranit Cohen-Ezra, Yana Davidov, Peretz Weiss, Tanya Bradichevski, Shlomo Segev, Omry Koren, Michal Safran, and Ziv Ben-Ari

Corresponding Author(s): Yelena Lapidot, Chaim Sheba Medical center

Review Timeline:

Submission Date:	February 18, 2020
Editorial Decision:	April 5, 2020
Revision Received:	April 28, 2020
Accepted:	May 26, 2020

Editor: Chaysavanh Manichanh

Reviewer(s): Disclosure of reviewer identity is with reference to reviewer comments included in decision letter(s). The following individuals involved in review of your submission have agreed to reveal their identity: Jiajia Ni (Reviewer #2)

Transaction Report:

DOI: <https://doi.org/10.1128/mSystems.00153-20>

April 5, 2020

Dr. Yelena Lapidot
Chaim Sheba Medical center
Liver Disease Center
Chaim Sheba Medical center
Tel Hashomer
Ramat Gan
Israel

Re: mSystems00153-20 (Alterations in the gut microbiome in the progression of cirrhosis to hepatocellular carcinoma)

Dear Dr. Yelena Lapidot:

Thank you for submitting your manuscript to System. Your submission has now been assessed by external advisers. We would like to invite you to REVISE your paper in light of the reviewers' comments below. Please carefully address all the issues raised in the comments.

To submit your modified manuscript, log onto the eJP submission site at <https://msystems.msubmit.net/cgi-bin/main.plex>. If you cannot remember your password, click the "Can't remember your password?" link and follow the instructions on the screen. Go to Author Tasks and click the appropriate manuscript title to begin the resubmission process. The information that you entered when you first submitted the paper will be displayed. Please update the information as necessary. Provide (1) point-by-point responses to the issues raised by the reviewers as file type "Response to Reviewers," not in your cover letter, and (2) a PDF file that indicates the changes from the original submission (by highlighting or underlining the changes) as file type "Marked Up Manuscript - For Review Only."

Due to the SARS-CoV-2 pandemic, our typical 60 day deadline for revisions will not be applied. I hope that you will be able to submit a revised manuscript soon, but want to reassure you that the journal will be flexible in terms of timing, particularly if experimental revisions are needed. When you are ready to resubmit, please know that our staff and Editors are working remotely and handling submissions without delay. If you do not wish to modify the manuscript and prefer to submit it to another journal, please notify me of your decision immediately so that the manuscript may be formally withdrawn from consideration by mSystems.

To avoid unnecessary delay in publication should your modified manuscript be accepted, it is important that all elements you upload meet the technical requirements for production. I strongly recommend that you check your digital images using the Rapid Inspector tool at <http://rapidinspector.cadmus.com/RapidInspector/zmw/>.

Sincerely,

Chaysavanh Manichanh

Editor, mSystems

Journals Department
Reviewer comments:

Reviewer #1 (Comments for the Author):

In the present study, Yelena L. et al aimed to elucidate alterations in the gut microbiome in the progression of cirrhosis to HCC. The statistical analysis has been performed appropriately and rigorously, and the results and conclusions are consistent with the main objective. The study is very interesting, but for this reviewer only there are a few comments:

- In the group of patients with cirrhosis, patients with NAFLD and HCV are included, but the number of each of the subgroups is not defined. I think this data is important because the initial microbial composition due to the cause is already very important, although the objective is to define the microbiota in the HCC group.
- In the same context of the previous comment, I think figure S3 is important enough not to be supplementary, because is the basis of the cirrhosis group
- In the text, the different sections of the figures should be better defined when explaining the results

Reviewer #2 (Comments for the Author):

In this study, the authors compared the differences of gut microbiota compositions between the patients with HCC-cirrhosis and cirrhosis, and healthy controls. Although they proposed a very important question that "the factors influencing progression from cirrhosis to HCC remain largely unknown", their results did not answer the question. I know it is very hard to carry out clinical experiments and it is difficult to control completely. However, just comparing the differences of gut microbiota compositions between the patients with HCC-cirrhosis and cirrhosis, and healthy controls cannot provide valuable information to solve the problem, despite the authors provided dietary information from the food frequency questionnaires.

In addition, since HCV is an important factor that causes cirrhosis and HCC, the authors should distinguish whether the HCC patients have HCV at the same time, as well as the cirrhosis patients. If the authors want to provide optional HCC diagnostic markers, they should distinguish the stages of HCC, the differences of gut microbiota composition in early HCC patients is more valuable for early diagnosis.

Therefore, I believe the authors should reorganize their scientific question and discussion section.

There are many minor errors also should be revised as follows (Because I am not a native English speaker, I can't evaluate the language of the manuscript, although I feel there are some grammar problems),

Line 3-5: I think the "." after the authors family name should be deleted. Please confirm again.

Line 109-110: In the 30 cirrhotic subjects with HCC, how many patients were NAFLD or HCV related?

Line 111-112: Please add a reference or more details for diagnosing cirrhosis.

Line 115-116: Please add a reference or more details for diagnosing HCC. And I suggest the authors should stage the patients with HCC.

Line 134: What were saliva samples used for?

Line 155 and line 199: Please change "OTU" to "feature" as QIIME2 don't longer use the concept of "OTU".

Line 156: Why the author used unweighted UniFrac method but not weighted UniFrac method to indicate the beta-diversity? If considering the relative abundance of each facture, weighted UniFrac method was maybe more suitable.

Line 159: Please change "qvalue" to "q value".

Line 173: Which version of Greengenes database was used? 13.5?

Line 186-187: I think the sentence "All authorsapproved the final manuscript" should appear in the author's contribution section.

Line 240: Please add a "." after the "(Rho = 0.52, p-value = 0.001)".

Line 254: Please define the "HC".

Line 390: The format of references is very confusing, so it is suggested to modify it carefully according to the format requirements of the journal.

Page 27-41: The order of supplemental materials is disordered, please reorder the materials.

20/04/20

Response to reviewers: mSystems00153-20 “Alterations in the gut microbiome in the progression of cirrhosis to hepatocellular carcinoma”.

We thank the reviewers for reviewing our manuscript and their suggestions for improving our paper. In addition, we also thank the editorial team for providing us with the opportunity to revise our manuscript.

Please find below our response to the reviews:

Reviewer #1 (Comments for the Author):

In the present study, Yelena L. et al aimed to elucidate alterations in the gut microbiome in the progression of cirrhosis to HCC. The statistical analysis has been performed appropriately and rigorously, and the results and conclusions are consistent with the main objective. The study is very interesting, but for this reviewer only there are a few comments:

- In the group of patients with cirrhosis, patients with NAFLD and HCV are included, but the number of each of the subgroups is not defined. I think this data is important because the initial microbial composition due to the cause is already very important, although the objective is to define the microbiota in the HCC group.

Answer: In light of the reviewer comment we have added the distribution of cirrhosis etiologies in both groups (cirrhosis and HCC-cirrhosis), in the methods section: “Study groups included 38 patients with NAFLD (n=19) or Hepatitis C (HCV) (n=19) related liver cirrhosis, 30 cirrhotic subjects with a first diagnosed early stage HCC (NAFLD related HCC-Cirrhosis (n=14) and HCV related HCC-Cirrhosis (n=16)) and 27 age- and BMI-matched healthy controls.” Furthermore, in the first paragraph of the results we have referred to the

statistical differences between the patients groups “There were no significant differences in cirrhosis etiology, severity, mean age and BMI between the groups (Table 1)”. Table 1 includes the detailed distribution of disease etiologies as well.

- In the same context of the previous comment, I think figure S3 is important enough not to be supplementary, because is the basis of the cirrhosis group.

Answer: As suggested by the reviewer, we have merged figure S3 into figure 1 (Fig1 C and D) and we have added the basis of the HCC-cirrhosis group etiologies as well (Fig 1 E and F).

- In the text, the different sections of the figures should be better defined when explaining the results

Answer: We have added clear definitions of figure sections.

Reviewer #2 (Comments for the Author):

In this study, the authors compared the differences of gut microbiota compositions between the patients with HCC-cirrhosis and cirrhosis, and healthy controls. Although they proposed a very important question that "the factors influencing progression from cirrhosis to HCC remain largely unknown", their results did not answer the question. I know it is very hard to carry out clinical experiments and it is difficult to control completely. However, just comparing the differences of gut microbiota compositions between the patients with HCC-cirrhosis and cirrhosis, and healthy controls cannot provide valuable information to solve the problem, despite the authors provided dietary information from the food frequency questionnaires.

In addition, since HCV is an important factor that causes cirrhosis and HCC, the authors should distinguish whether the HCC patients have HCV at the same time, as well as the cirrhosis patients.

Answer: As mentioned in the methods section, all patients with HCV-cirrhosis were recruited prior to HCV eradication.

If the authors want to provide optional HCC diagnostic markers, they should distinguish the stages of HCC, the differences of gut microbiota composition in early HCC patients is more valuable for early diagnosis.

Answer: As stated by the reviewer, we also believe that the alterations of gut microbiome composition in early HCC patients is of great value for early diagnosis, therefore for this study we have recruited only subjects with a first diagnosis of an early stage HCC, prior to treatment. Furthermore, in order to clarify this issue, we have added the following sentence to the method section “Study groups included 38 patients with NAFLD (n=19) or Hepatitis C (HCV) (n=19) related liver cirrhosis, 30 cirrhotic subjects **first diagnosed for early stage HCC, prior to treatment**”.

Therefore, I believe the authors should reorganize their scientific question and discussion section.

Answer: Based of the reviewer suggestions, we have reorganized the scientific question, focusing on **characterization** of the fecal microbiome in cirrhosis and HCC-cirrhosis, with specific evaluation of etiology, progression and environmental factors (BMI and diet). We have also added a paragraph that refers to the study limitations in the discussion section:

"Our findings demonstrate the potential of fecal microbes as tools for noninvasive diagnosis

or microbiome oriented interventions in HCC-cirrhosis. However, this study focused on characterization of the 16S rRNA bacterial alterations and the potential effects of etiology, cirrhosis progression and factors as BMI and diet on these patients bacterial composition, thus causality remains an open question. Nevertheless, this study showed sufficient power for detecting significant microbial alterations and successfully illuminating the effect of environmental factors on the gut microbiome of patients with HCC-cirrhosis. In the future, there will be a need for further large-scale, multi-racial, metagenomic cohorts, that will include a careful evaluation of environmental factors, including dietary habits and their effect on patients with HCC-cirrhosis. Moreover, there is a need for intervention studies, both in animal models and in humans, focusing on fecal microbiome modulation in cirrhosis and HCC-cirrhosis. These studies will allow further understanding of the roles of altered bacteria in the progression of cirrhosis to HCC."

There are many minor errors also should be revised as follows (Because I am not a native English speaker, I can't evaluate the language of the manuscript, although I feel there are some grammar problems),

Line 3-5: I think the "." after the authors family name should be deleted. Please confirm again.

Answer: Corrected.

Line 109-110: In the 30 cirrhotic subjects with HCC, how many patients were NAFLD or HCV related?

Answer: In agreement with the reviewer comments, we have specified the etiology of cirrhosis in both patients groups (cirrhosis without HCC and HCC-cirrhosis). As mentioned in the manuscript we included patients with NAFLD and HCV related cirrhosis in both

groups (table 1). Moreover, we have added to figure 1 the association of bacterial diversity and composition with the distribution of cirrhosis etiologies in cirrhosis and HCC-cirrhosis groups.

Line 111-112: Please add a reference or more details for diagnosing cirrhosis.

Answer: We have added a reference for the diagnosis of cirrhosis.

9. Procopet B, Berzigotti A. 2017. Diagnosis of cirrhosis and portal hypertension: imaging, non-invasive markers of fibrosis and liver biopsy. *Gastroenterol Rep* 5:79–89.

Line 115-116: Please add a reference or more details for diagnosing HCC. And I suggest the authors should stage the patients with HCC.

Answer: We have added a reference for the diagnosis of HCC. In addition, we clarified that in this study, only newly diagnosed HCC patients were recruited, prior to receiving any treatment and with early stage HCC.

10. Marrero JA, Kulik LM, Sirlin CB, Zhu AX, Finn RS, Abecassis MM, Roberts LR, Heimbach JK. 2018. Diagnosis, Staging, and Management of Hepatocellular Carcinoma: 2018 Practice Guidance by the American Association for the Study of Liver Diseases Purpose and Scope 68.

Line 134: What were saliva samples used for?

Answer: For this analysis we have used fecal samples only, this line was corrected.

Line 155 and line 199: Please change "OTU" to "feature" as QIIME2 don't longer use the concept of "OTU".

Answer: Corrected.

Line 156: Why the author used unweighted UniFrac method but not weighted UniFrac method to indicate the beta-diversity? If considering the relative abundance of each feature, weighted UniFrac method was maybe more suitable.

Answer: Following the reviewers suggestion, we have also added an analysis of weighted UniFrac, at the same sequencing depth (Figure S1). The use of weighted and unweighted measures of β diversity can reveal different factors influencing the microbial communities and together can illuminate our understanding of bacterial composition alterations.

Unweighted UniFrac is suited for detection of differences in the presence or absence of lineages of bacteria in different communities and is more sensitive to differences in low-abundance features, whereas weighted UniFrac is suited for detection of differences in microbial abundances that may arise in cases such as transient changes due to nutrient availability (Lozupone CA, Hamady M, Kelley ST, Knight R. 2007. Quantitative and qualitative β diversity measures lead to different insights into factors that structure microbial communities. *Appl Environ Microbiol*. American Society for Microbiology).

In our study, unweighted UniFrac revealed a clear association between fecal community composition and liver disease (cirrhosis and HCC-cirrhosis as compared to controls, p value = 0.004 and p value = 0.016 correspondingly), while weighted UniFrac resulted in weaker association (Cirrhosis as compared to controls p value = 0.243, HCC-cirrhosis as compared to controls p value = 0.263), implying that the bacterial alterations occur in low abundance features. The effect of low abundant features was depleted in the calculation of weighted UniFrac. We have added the calculation of weighted UniFrac to the results, including figure S1 and elaborated the meaning of these results in the discussion section.

Figure S1: Weighted phylogenetic β -diversity analysis. Box plots of phylogenetic β -diversity measured by Weighted UniFrac distance matrix, displaying no significant differences between study groups (Cirrhosis as compared to controls p value = 0.243, HCC-cirrhosis as compared to controls p value = 0.263, cirrhosis compared to HCC-cirrhosis p value = 0.117).

Line 159: Please change "qvalue" to "q value".

Answer: Corrected.

Line 173: Which version of Greengenes database was used? 13.5?

Answer: We have used version 13.8 of Greengenes, and included this information in the methods section.

Line 186-187: I think the sentence "All authorsapproved the final manuscript" should appear in the author's contribution section.

Answer: Corrected.

Line 240: Please add a "." after the "(Rho = 0.52, p-value = 0.001)".

Answer: Corrected.

Line 254: Please define the "HC".

Answer: This was an abbreviation for healthy controls. We have corrected this in the manuscript.

Line 390: The format of references is very confusing, so it is suggested to modify it carefully according to the format requirements of the journal.

Answer: The references were carefully reformed according to the American society of microbiology guidelines.

Page 27-41: The order of supplemental materials is disordered, please reorder the materials.

Answer: We have reorganized the supplementary materials.

May 26, 2020

Dr. Yelena Lapidot
Chaim Sheba Medical center
Liver Disease Center
Chaim Sheba Medical center
Tel Hashomer
Ramat Gan
Israel

Re: mSystems00153-20R1 (Alterations in the gut microbiome in the progression of cirrhosis to hepatocellular carcinoma)

Dear Dr. Yelena Lapidot:

Thank you for submitting your revision, which has now been re-assessed by external advisers, who have recommended publication. I am forwarding it to the ASM Journals Department for publication. For your reference, ASM Journals' address is given below. Before it can be scheduled for publication, your manuscript will be checked by the mSystems senior production editor, Ellie Ghatineh, to make sure that all elements meet the technical requirements for publication. She will contact you if anything needs to be revised before copyediting and production can begin. Otherwise, you will be notified when your proofs are ready to be viewed.

Sincerely,

Chaysavanh Manichanh
Editor, mSystems

Journals Department
Table S4: Accept
Table S3: Accept
Fig. S2: Accept
Fig. S3: Accept
Fig. S1: Accept
Table S6: Accept
Fig. S4: Accept
Table S1: Accept
Table S2: Accept
Table S5: Accept